# GEO-Bench:
# Toward Foundation Models for Earth Monitoring

**Alexandre Lacoste**[*1]     **Nils Lehmann**[*2]     Pau Rodriguez[1]     Evan David Sherwin[3]

Hannah Kerner[4]     Björn Lütjens[5]     Jeremy Irvin[3]     David Dao[6]

Hamed Alemohammad[7]     Alexandre Drouin[1,8]     Mehmet Gunturkun[1]     Gabriel Huang[1,9]

David Vazquez[1]     Dava Newman[5]     Yoshua Bengio[8,9]     Stefano Ermon[3]

Xiao Xiang Zhu[2]

[1] ServiceNow Research     [2] Technical University of Munich     [3] Stanford University
[4] Arizona State University     [5] MIT     [6] ETH Zurich     [7] Clark University
[8] Mila-Quebec     [9] University of Montreal

## Abstract

Recent progress in self-supervision has shown that pre-training large neural networks on vast amounts of unsupervised data can lead to substantial increases in generalization to downstream tasks. Such models, recently coined *foundation models*, have been transformational to the field of natural language processing. Variants have also been proposed for image data, but their applicability to remote sensing tasks is limited. To stimulate the development of foundation models for Earth monitoring, we propose a benchmark comprised of six classification and six segmentation tasks, which were carefully curated and adapted to be both relevant to the field and well-suited for model evaluation. We accompany this benchmark with a robust methodology for evaluating models and reporting aggregated results to enable a reliable assessment of progress. Finally, we report results for 20 baselines to gain information about the performance of existing models. We believe that this benchmark will be a driver of progress across a variety of Earth monitoring tasks.

## 1   Introduction

Earth monitoring with machine learning-based methods plays an increasing role in climate change mitigation and adaptation as well as climate science [57]. Related applications include methane source detection [61, 16], forest carbon quantification [44], extreme weather prediction [49], and crop monitoring [34, 14]. Across many of these applications, pre-trained models (e.g., a ResNet trained on ImageNet) have been used to increase generalisation performance. Improvement of the pre-trained models has been shown to reduce the need for large labelled datasets in some contexts [11] and can improve model generalisation outside of the training distribution [28]. Recent studies exploring the scaling of such pre-trained models found that increasing the size of an unsupervised (or weakly supervised) dataset as well as properly scaling the model led to an even greater increase in performance under various metrics [33, 55].

While the training of such large-scale models is usually reserved for industrial research groups with very large computer clusters, the publication of pre-trained models creates vast opportunities for the entire research and technology community (including communities of domain experts outside of machine learning). These large pre-trained models were recently coined as *foundation models* [6] as they might serve as foundations for sub-fields of machine learning. Specifically, the publication of large pre-trained models like BERT [15] and GPT-3 [7] led to a paradigm shift in the field of natural language processing (NLP). This inspired a similar shift in the field of computer vision with the release of models like CLIP [55] and DINO [9]. While CLIP performs well on various types of vision tasks, it still under-performs

37th Conference on Neural Information Processing Systems (NeurIPS 2023) Track on Datasets and Benchmarks.

on Earth monitoring tasks [55]. This is not surprising as it is trained mainly on RGB images taken from a ground perspective at a single point in time.

While there are many similarities between Earth observation datasets and typical ML image datasets, there are also many important differences to consider when designing effective ML models. Earth observation images are taken from an overhead rather than ground perspective, usually from a fixed distance from the Earth's surface (defined by a satellite's orbit). The satellite revisits provide a temporal axis that is sometimes irregular (e.g., a few times per year) or regular (e.g., every five days) with cloud coverage causing spurious occlusions. Images are acquired with sensors containing multiple spectral bands (e.g., thirteen for Sentinel-2), or even with different kinds of sensors, e.g., synthetic aperture radar (SAR), which can penetrate cloud coverage. Moreover, the GPS coordinates and timestamp of each acquisition offer the opportunity to combine data from multiple sources, e.g., weather data, semantic maps, and elevation. This leads to a rich multi-modal signal with potentially missing information that can be inferred from other elements of the signal. There are currently petabytes of accessible satellite datasets containing images of the Earth under various modalities from the present day to as far back as the 1960s. Distilling this large amount of information into pre-trained models of various sizes offers the opportunity to redistribute this information and make it accessible to various labs for increasing the performances on a large range of downstream tasks.

The fundamental goal of these large pre-trained models is to improve generalization performance on downstream tasks. Hence, to support the machine learning community in producing better pre-trained models, it is crucial to provide a benchmark with a wide variety of downstream tasks, covering a range of modalities and dataset shapes that are likely to be encountered in practice. At the moment, existing works on pre-training models from earth observations e.g., [13, 46, 69], evaluate on different sets of downstream tasks, making it impossible to directly compare performance. Moreover, the set of tasks is often narrow in terms of diversity and the statistical methodologies do not adequately report the uncertainties in the evaluation.

The present work aims to fill this void by providing a wide range of tasks across various countries with various modalities of sensors. Also, the transformed versions of the datasets are smaller than their original form, and all results can be replicated on single GPUs. This increases accessibility to research labs with limited resources and reduces overall energy consumption. Our proposed benchmark, GEO-Bench[1], is composed of six image classification and six semantic segmentation tasks, which were curated by domain experts to ensure their diversity and relevance toward sustainable development. We expect this contribution to:

- Stimulate and facilitate the development of foundation models for Earth monitoring
- Provide a systematic way of measuring the quality of models for better scientific progress
- Provide insights into which pre-trained models work best
- Potentially reduces negative impacts of foundation models through an open evaluation procedure.

In what follows, we start by discussing sources of data that can serve to train foundation models for earth monitoring (Sec. 2). We then present the details of GEO-Bench (Sec. 3) and how it can be used for the evaluation of foundation models (Sec. 4). Further, we review existing benchmark datasets for earth monitoring and discuss why GEO-Bench is complementary (Sec. 5). Finally, we present an extensive set of experiments, showing the performance of 20 state-of-the-art models on the benchmark to lay down reference points and to gain valuable information on existing pre-trained models (Sec. 6).

## 2    Remote sensing data for self-supervision

The development of foundation models does not typically rely on a specific dataset for the pre-training phase. The choice of data source is part of the design of the model, e.g., a very large corpus of text from the internet [50] or pairs of text associated with images from the web [55]. As such, we do not provide data for training foundation models with this benchmark. However, for completeness, we outline potential sources of Earth observation data that could be used for pre-training foundation models.

**Multispectral images with revisits**    Satellite data sources such as Sentinel-2 [20, 23] and Landsat 8 [66] provide images in multiple spectral bands with periodic revisits. This yields a four-dimensional array of structured data (longitude, latitude, wavelength, time) which can be used to perform various forms of self-supervision, e.g., predicting adjacent tiles [30] or contrasting the different seasons for the same region [46].

---

[1]https://zenodo.org/communities/geo-bench

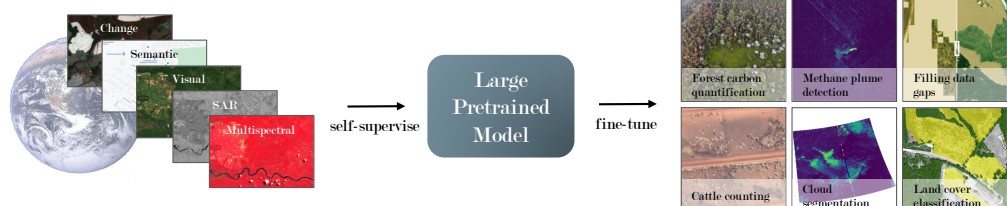

Figure 1: Foundation models encapsulate multimodal data streams through self-supervised training. The trained models can then be fine-tuned for a variety of climate-related remote sensing tasks. Image sources: quantification [44], detection [32], generation [43], counting [36], segmentation [75], and multi-class classification [51].

**Other sensors**    Synthetic Aperture Radar (SAR) and terrain elevation are also frequently available and can be matched to other sources of data through geolocalisation [54]. Such data are complementary to optical spectral bands and may encourage the model to learn higher-level semantic representations.

**Semantic data**    Through georeferencing, text-based data such as Wikipedia articles can be linked to satellite images [67]. It is also possible to join content from non-image data layers like OpenStreetMap [39]. By predicting or contrasting information from these sources, the model may learn useful and transferable semantic representations.

## 3   GEO-Bench

GEO-Bench is composed of 6 classification tasks and 6 segmentation tasks. Detailed characteristics are presented in Table 1, examples are depicted in Figure 2 and 3, and the spatial coverage on the world map is presented in Figure 8 (supplementary material). In what follows, we describe the procedure for collecting and transforming the datasets.

### 3.1   Design Principles

GEO-Bench was established by modifying and gathering geospatial datasets, adhering to principles that secure accessibility, usability, and effective model performance assessment across tasks.

**Ease of Use**    A fundamental goal was to create an accessible, simple-to-use benchmark, and a compact dataset assortment with code for loading the data in a consistent schema. A key aim was to harmonize data to reduce the engineering work needed to tailor pre-trained architectures, while maintaining sensor type and resolution diversity.

**Sector Experts and Steering Committee**    To align GEO-Bench with practical use-cases, we assembled a team of six sector experts from fields such as forestry and climate science. A steering committee of respected scientists guides high-level benchmark decisions, assuring relevance and impact.

**Diversity of Modalities**    The objective is to evaluate model adaptability to varied geospatial sensors. Thus, the benchmark encompasses multispectral, SAR, hyperspectral, elevation, and cloud probability modalities, with spatial resolutions from 0.1 to 30 m/pixel.

**Diversity of Tasks**    We ventured beyond image classification, incorporating object detection and semantic segmentation. To maintain *ease of use*, detection and counting tasks were transformed into semantic segmentation. This led to two task sets: six image classification tasks, and six semantic segmentation tasks [25, 38].

**Original Train, Validation, and Test Splits**    Original dataset splits were preserved when available; otherwise, we generated validation and test sets from the train set while ensuring no spatial overlap.

**Permissive License**    Most datasets needed to be adapted from their original form to satisfy the above criteria and be included in the benchmark. Hence, we include only datasets with permissive licenses.

**Classification**

| Name | Image Size | # Classes | Train | Val | Test | # Bands | RGB res | Sensors | Cite | License |
|---|---|---|---|---|---|---|---|---|---|---|
| m-bigearthnet | 120 x 120 | 43 | 20000 | 1000 | 1000 | 12 | 10.0 | Sentinel-2 | [64] | CDLA-P-1.0 |
| m-so2sat | 32 x 32 | 17 | 19992 | 986 | 986 | 18 | 10.0 | Sentinel-2 + Sentinel-1 | [76] | CC-BY-4.0 |
| m-brick-kiln | 64 x 64 | 2 | 15063 | 999 | 999 | 13 | 10.0 | Sentinel-2 | [37] | CC-BY-SA 4.0 |
| m-forestnet | 332 x 332 | 12 | 6464 | 989 | 993 | 6 | 15.0 | Landsat-8 | [29] | CC-BY-4.0 |
| m-eurosat | 64 x 64 | 10 | 2000 | 1000 | 1000 | 13 | 10.0 | Sentinel-2 | [27] | MIT |
| m-pv4ger | 320 x 320 | 2 | 11814 | 999 | 999 | 3 | 0.1 | RGB | [48] | MIT |

**Segmentation**

| Name | Image Size | # Classes | Train | Val | Test | # Bands | RGB res | Sensors | Cite | License |
|---|---|---|---|---|---|---|---|---|---|---|
| m-pv4ger-seg | 320 x 320 | 2 | 3000 | 403 | 403 | 3 | 0.1 | RGB | [48] | MIT |
| m-chesapeake-landcover | 256 x 256 | 7 | 3000 | 1000 | 1000 | 4 | 1.0 | RGBN | [56] | CDLA-P-1.0 |
| m-cashew-plantation | 256 x 256 | 7 | 1350 | 400 | 50 | 13 | 10.0 | Sentinel-2 | [74] | CC-BY-4.0 |
| m-SA-crop-type | 256 x 256 | 10 | 3000 | 1000 | 1000 | 13 | 10.0 | Sentinel-2 | link | CC-BY-4.0 |
| m-nz-cattle | 500 x 500 | 2 | 524 | 66 | 65 | 3 | 0.1 | RGB | [1] | CC-BY-4.0 |
| m-NeonTree | 400 x 400 | 2 | 270 | 94 | 93 | 5 | 0.1 | RGB + Hyperspectral + Elevation | [71] | CC0 1.0 |

Table 1: **GEO-Bench:** Characteristics of datasets in the benchmark. Since datasets are *modified*, we prepend their name with "m-" to distinguish them from the original dataset.

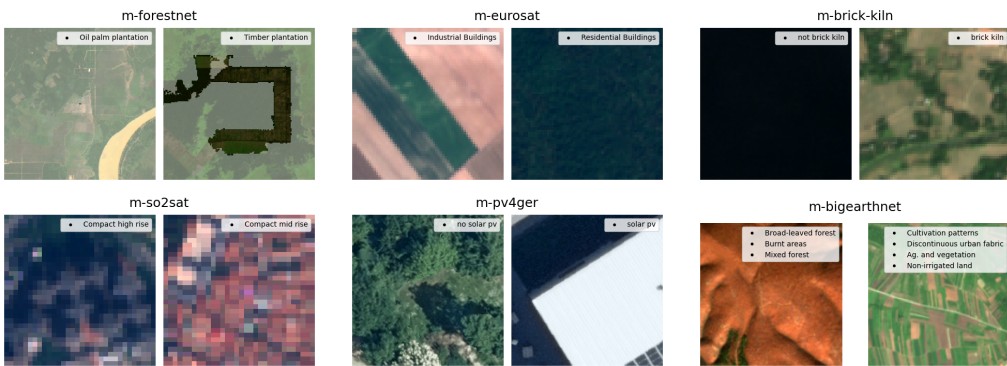

Figure 2: Representative samples of the **classification benchmark**.

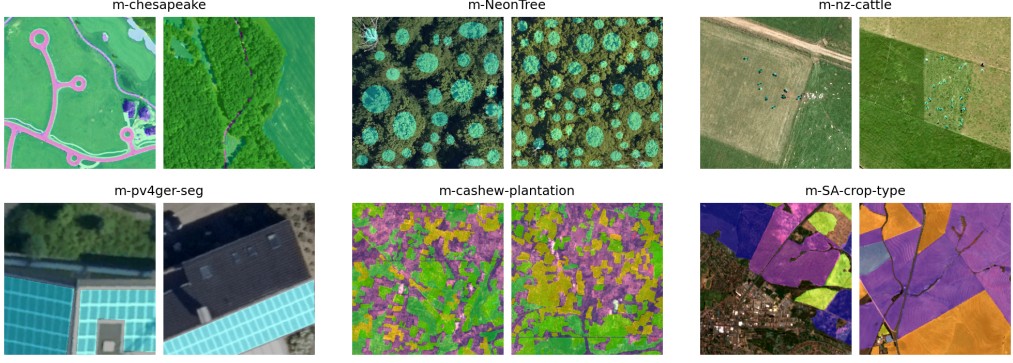

Figure 3: Representative samples of the **segmentation benchmark**.

## 3.2 Dataset Transformations

To produce a benchmark that complies with the design choices of Section 3.1, we applied the following transformations to each dataset. The procedure that was used to download and transform each dataset is fully documented and open-sourced in the GEO-Bench GitHub repository[2].

---

[2] https://github.com/ServiceNow/geo-bench

**Subsampling Large Datasets**    To be more representative of typical downstream tasks, where data is usually scarce, datasets larger than 20000 samples were randomly subsampled. Avoiding large downstream tasks also comes with other benefits:

- In Appendix A, we show that larger downstream datasets can decrease the ability to discriminate between two models that are similar in performance.
- Downstream tasks with very large training sets will not usually benefit from pre-training[3]. Hence they are less useful for our evaluation purpose.
- A smaller benchmark is faster to download, yields results quicker and requires less energy for computation.
- We can increase the variety of experiments and the number of seeds to improve the knowledge gained from experiments.

**Removing Class Imbalance**    We randomly subsampled large classes to have near-uniform class sizes across datasets. This was done to prevent users of the benchmark from increasing their score by using clever class imbalance techniques instead of making progress on better pre-trained models. While good performance on highly imbalanced (long tail of classes) datasets would be a desired property of a pre-trained model, we have not found a good dataset containing a large number of classes.

## 4    Using The Benchmark

**Fine Tuning**    In the self-supervised learning literature, it is common to use the pre-trained model to encode a fixed representation of each image in the dataset and learn to classify images based on this representation [30]. While this works relatively well, this method highly depends on the pre-training task as it may not learn to encode information that is important for the downstream task [65, 53]. In practice, fine-tuning the pre-trained model often mitigates this issue and is known to frequently yield a much higher generalization performance than a model trained from random weights [46, 11]. Since this is more representative of practical usage, we encourage users of the benchmark to report the results of fine-tuned models. On the other hand, we do not discourage users from also reporting results with fixed backbones (pre-trained weights) as this can provide valuable information about the pre-trained model. In all cases, we ask users to report their fine-tuning methodology with enough details for reproducibility.

**Hyperparameter Tuning**    Deep learning algorithms often require the adjustment of hyperparameters, especially when an architecture is fine-tuned on a small dataset. For this reason, we recommend adjusting hyperparameters, but within a maximum budget of 16 trials per task[4]. Early stopping based on validation metrics is also recommended.

**Data Augmentation**    Data augmentation plays a crucial role in the training of deep learning models, especially with small training datasets. Hence, we consider it to be part of the fine-tuning process. As a guideline, we propose limiting the augmentations to $90°$ rotations and vertical and horizontal flips[5]. On the other hand, we also encourage users to study what are the best data augmentations for remote sensing as this could lead to useful findings for practitioners and the benchmark is well-suited for evaluating such findings.

**Toolbox**    To facilitate the usage of the benchmark, we provide a collection of tools for various parts of the experimental pipeline as part of the open-sourced codebase[6]. This includes tools for loading datasets and visualising results. We also provide tools based on PyTorch-Lightning [24] to facilitate model training.

### 4.1    Reporting Results

For reliable and comparable results across different publications, we recommend that users follow this procedure to report results. The aim is to report results on individual tasks as well as aggregated across all tasks, with reliable confidence intervals (inspired by [2]). Code is provided to generate figures based on raw results.

---

[3]From Bayes rule, we know that the influence of the prior (pre-trained model) decreases as the size of the training data increases.

[4]While 16 is fairly small, we believe it's enough to adjust sensitive hyperparameters such as learning rate. Also, this favours models that are less sensitive to hyperparameter tuning.

[5]Random crop and resize are also common in vision, but in remote sensing, this reduces the spatial resolution, which is often crucial for high performances.

[6]https://github.com/ServiceNow/geo-bench

**Random Seeds**  As demonstrated in [2], 3-5 seeds are not enough to obtain reliable confidence intervals. Since pre-training and hyperparameter search are usually the computational bottlenecks, we recommend retraining the selected hyperparameter configuration for at least 10 different seeds.

**Interquartile Mean (IQM)**  We recommend using IQM. This metric removes the outliers by trimming the 25% highest values as well as the 25% lowest value and computing the average of the remaining values. The resulting finite sample estimator is less biased than the median and has less variance than the mean, often resulting in smaller confidence intervals [2].

**Normalising Results**  To aggregate performance metrics across multiple tasks, one must first normalise their values. A common approach consists of applying a linear transformation based on reference points [4]. As such, we propose to use the lowest and highest metric values achieved by a set of strong baselines (see Sec. 6) as *official reference points*. For each individual task, we scale the results such that the maximum score is 1 and the lowest one is 0. Hence, if a future model were to achieve a score superior to 1, it would imply that progress is being made on the benchmark. All reference points will be published alongside the benchmark.

**Bootstrapping**  To quantify uncertainty over observed IQMs, we use bootstrapping [21]. That is, we sample $n$ times, with replacement, the results from training with $n$ different seeds, and we compute IQM. Repeating this procedure $n\!=\!1000$ times provides a distribution over IQM results, from which confidence intervals can be extracted.

**Aggregated Results**  After normalizing the results we simply compute IQM across all datasets and all results of a given model. For confidence intervals, we use *stratified bootstrap*, where seeds are sampled with replacement *individually* for each dataset, but IQM is computed across all datasets.

**Displaying the results**  In Figure 4, we show how to compactly display results from a wide range of baselines across the benchmark as well as aggregated results and statistical uncertainties. In Figure 5, we display the results for a growing training set size (with fixed validation and test set). This compactly reports the results of thousands of experiments.

**Publishing the results**  We ask experimenters to publish the results of all seeds on all datasets for all models as a CSV file along with the open-sourced code of their experiments. This will allow future authors to incorporate existing results in their comparison figures.

## 5  Related Works

**SustainBench**  consists of 15 public datasets covering 7 sustainable development goals [73]. Seven of these datasets are two-dimensional remote sensing. It includes a public leaderboard for tracking model performance. A featured task is the Brick Kiln classification, selected for its georeferenced, high-quality ground truth labels. SustainBench's purpose is monitoring progress in specified tasks, thus comprising a diverse set of datasets. It doesn't aim for solution under a single framework or aggregate result tracking.

**TorchGeo**  is a Python library designed to streamline the integration of remote sensing datasets into the PyTorch deep learning ecosystem [62]. TorchGeo currently features data loaders for 52 publicly available datasets of satellite, aerial, and drone imagery for classification, regression, change detection, semantic segmentation, instance segmentation, and object detection tasks. Our benchmark directly interfaces with TorchGeo and uses its data loaders for several datasets included in the benchmark.

**EarthNets**  is a concurrently developed platform to evaluate deep learning methods on remote sensing datasets [72]. In their methodology, they analyse the metadata of 400 publicly available remote sensing datasets. Using meta-information such as the number of samples, the size of each sample, and the type of annotations, they analyse the correlation between each dataset and identify a variety of clusters. Based on this analysis, they recommend two classification, two segmentation, and two detection datasets for benchmarking. In contrast, we provide a collection of 12 datasets and we propose a robust methodology for aggregating results and reporting statistical uncertainties of the evaluation process.

**AiTLAS**  recently proposed a benchmark of 22 classification datasets[17], 3 of which intersect with our classification benchmark. They proposed a standardised version of train, valid, test splits for existing datasets as well as a fine-tuning procedure. By leveraging the overlap of labels across datasets, they also

provide a more accurate test metric for real-world applications. Experiments are conducted using 10 different model families across the 22 datasets, using RGB images as input.

# 6 Experiments

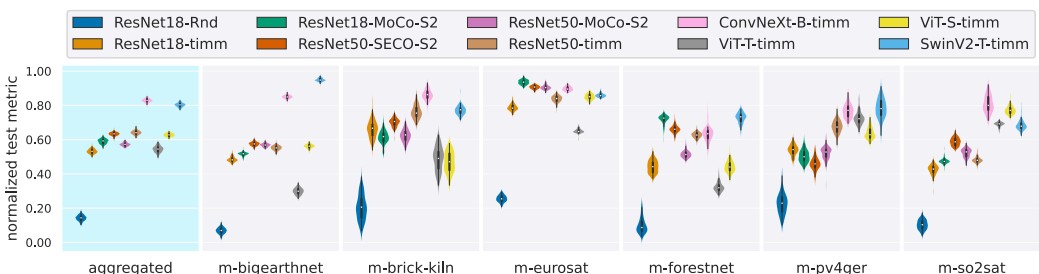

Figure 4: **Classification Benchmark RGB Only:** Normalised accuracies of various baselines (higher is better). Violin plots are obtained from bootstrap samples of normalized IQM (Section 4.1). The left plot reports the average across all tasks.

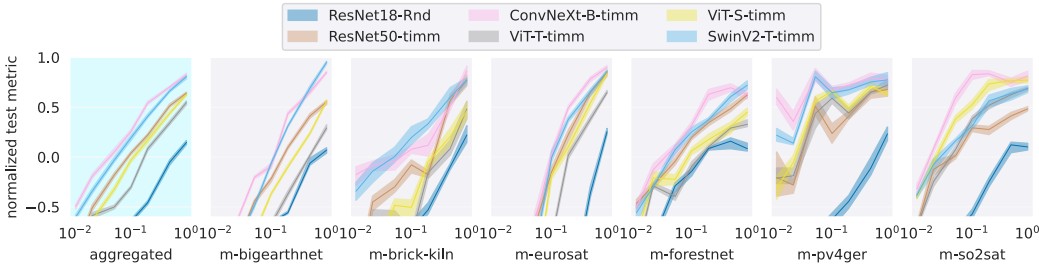

Figure 5: **Classification vs Train Size:** Normalised accuracies of a subset of the baselines on Classification benchmark with a growing size of the training set. The shaded region represents an 80% confidence interval, obtained from bootstrap samples of normalized IQM (Section 4.1). The left plot reports the average across all tasks.

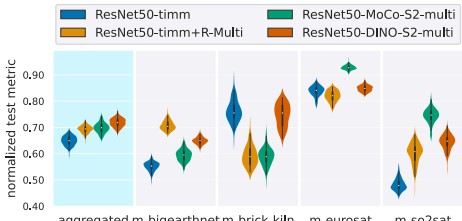

Figure 6: **Effect of Multispectral with ResNet50**. Only Sentinel-2 tasks are reported. Normalised accuracies (Sec 4.1).

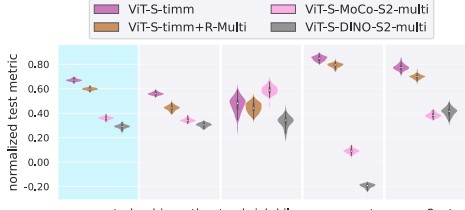

Figure 7: **Effect of Multispectral with ViT-S**. Only Sentinel-2 tasks are reported. Normalised accuracies (Sec 4.1).

In this section, we provide a range of baselines for the classification and segmentation benchmarks. These will serve as reference points for future evaluation[7]. We also seek to answer the following questions:

- Which new architecture performs best on remote sensing data (Section 6.2.2)?
- What is the effect of training set size on the performance of each model (Section 6.2.3)?
- Can we leverage multispectral channels to improve performance (Section 6.2.4)?
- Are smaller datasets better at discriminating the performance of different models (Section A.5)?

---

[7]We recall that all datasets have been modified from their original version. Hence, our results are not directly comparable to other published results.

### 6.1 Protocol

For each model, we replaced the last layer with a randomly initialised layer of the appropriate shape for the task at hand. We use different learning rates for the last layer (which starts from random weights) and for the backbone (which starts from pre-trained weights). The best learning rates were selected using the highest accuracy or Intersection over Union (IoU) on the validation set over 16 trials [8]. After choosing the hyperparameters, we repeated the training for 10 seeds. To minimize overfitting, we selected the best time step using accuracy (or IoU) on the validation set and we reported the test metrics at the chosen time step. We use AdamW [42] to train convolution architectures and SGD to train transformer architectures.

### 6.2 Classification

#### 6.2.1 Baselines Naming Schema

Each baseline name starts with the corresponding architecture: **ResNet18 and ResNet50:** standard ResNet architectures [26]; **ConvNeXt-B:** the base architecture of ConvNeXt [41]; **ViT-T and ViT-S:** ViT architectures [19] of size tiny and small respectively; **SwinV2-T:** a SwinV2-tiny architecture [40];

Then, keywords provide details about the training procedure: **SeCo:** a ResNet50 model trained on Sentinel 2 data with temporal contrastive loss across seasons [46]; **MoCo-S2 and DINO-S2:** model trained with self-supervision on Sentinel data [70] (RGB and Multispectral pre-trained weights); **Rnd:** weights are randomly initialised; **timm:** pre-trained weights are obtained from the timm library, usually from training on ImageNet; **+R-Multi:** we manually augment an RGB architecture by randomly initialising the weights of the missing channels in the 1st layer; **multi:** the pre-trained model has multispectral channels.

#### 6.2.2 Comparing Baselines on RGB only

In Figure 4, we report bootstrapped IQM of the normalized accuracy (Sec 4.1) for the six datasets of the classification benchmark, as well as aggregated results[9]. In this first experiment, all models can only see the RGB channels.

These results offer valuable information across 10 common baselines in the literature. We denote the outstanding performance of ConvNext and SwinV2 compared to other models. It is by a large margin the best models in aggregated results and almost systematically outperforms all models on all datasets. We can also observe the large difference between Scratch ResNet18 and ResNet18 on all datasets. This highlights the importance of using a pre-trained model. Also, perhaps disappointingly, the existing model pre-trained on remote sensing data does not exhibit any improvement compared to their timm pre-trained weights, i.e., ResNet18-MoCo-S2, ResNet50-MoCo-S2, and ResNet50-SeCo-S2 are all comparable to ResNet18 on the aggregated performance. On the other hand, in Section 6.2.4, we see that ResNet50-MoCo-S2-multi can leverage multispectral data to slightly surpass ResNet50-timm.

Another insight that can be gained from these results is how useful a dataset is at discriminating baselines, i.e., a dataset where most baselines perform equally would have limited utility in our benchmark. To this end, we had to discard GeoLifeClef 2022 [12] as all models were performing equally badly[10]. m-eurosat also offers limited discriminativity as most models obtain very high accuracy (see Figure 9). To make this dataset harder, we subsample down to 2000 training samples. We can now see that smaller models tend to perform better on this dataset, but the discriminativity remains fairly low.

#### 6.2.3 Accuracy vs training set size

As part of the benchmark, we also provide official subsets of the training sets with train ratios of (0.01, 0.02, 0.05, 0.1, 0.2, 0.5, 1) [11].

---

[8]The range of selected learning rates is different for each model and is selected based on early experiments, see appendix for details.

[9]We note that the variance of the results represents the uncertainty of the mean (IQM) which is significantly smaller than the variance of the raw seeds presented in Figure 9 in Appendix.

[10]We suspect this dataset to have high aleatoric uncertainties.

[11]Reporting results on all 7 subsets increases the number of experiments by 7x. However, in Figure 13 (see Appendix), we show that the convergence time is proportional to the training set size. This means that training on all seven subsets takes on average about 1.88 times longer than just training on the full training set.

Figure 5 depicts a different perspective on the models. First, we can observe the noise due to the hyperparameter selection process that is not accounted for by repeating 10 seeds with fixed hyperparameters. Also, we see that ConvNeXt often becomes better than SwinV2 as the training set decreases. This coincides with the common observations that transformer architectures tend to be more data-hungry, but also tend to outperform convolution architectures in the high data regime [18]. We note also, that ConvNeXt-B-timm only requires 2% of the training set to obtain aggregated performances comparable to that of ResNet18-Rnd. This impressive factor of 50x on data efficiency highlights the importance of developing new architectures and new pre-training methods. Finally, we can observe an increase in the discriminativity of the datasets as the training set decreases, specifically for m-eurosat, when the task becomes more difficult, the strong baselines stand out even more. The discriminativity of datasets is further studied in Section A.5.

### 6.2.4 Leveraging Multispectral Information

We now study the effect of leveraging multispectral information during the pre-training phase and during the fine-tuning phase. We do so by fixing the backbone to either ResNet50 (Fig. 6) or ViT-S (Fig. 7) and exploring various weight initialisation schema. Since we could only find pre-trained models for Sentinel-2, we limit this experiment to the four datasets satisfying this criterion.

We found that using a model pre-trained on RGB-only (timm pre-trained) and augmenting the architecture by randomly initialising the weights of the missing channels in the first layer (+RMulti) does not lead to systematic improvement. Moreover, the fine-tuning time is largely extended since we have to wait until the newly initialised weights on the first layer fully converge. On the other hand, the ResNet50 pre-trained on Sentinel-2 using DINO or MoCo [70] leads to a modest performance increase on average. When looking at ViT-S (Fig. 7), incorporating multi-spectral only leads to a systematic performance decrease.

### 6.3 Segmentation

We defer experiments on the Segmentation benchmark to Appendix A.3, where we provide experiments on six baselines (ResNet18, ResNet50, ResNet101) × (U-Net, DeepLabV3) with pre-trained weights provided by the timm library. While ResNet101-DeepLabV3 performs best in aggregate, it still underperforms on some datasets.

### 6.4 Resource Usage

See Appendix A.6 for detailed resource usage of each algorithm evaluated in this section. We report the number of parameters, memory usage, the time required for a forward pass, and the convergence time for fine-tuning on downstream tasks. While memory footprint can increase by a factor of 4x for a model like SwinV2 and ConvNeXt-B compared to ResNet50, their forward pass is only twice as slow.

## 7 Conclusion

We developed a new benchmark for evaluating pre-trained models on remote sensing downstream tasks. This involves adapting a variety of remote sensing datasets to a more conventional machine learning pipeline and providing code for fine-tuning and evaluating individual tasks. We expect that this benchmark will stimulate the development of new foundation models that could lead to better generalization on a variety of earth monitoring downstream tasks and could open up opportunities for new applications.

**Limitations** Our benchmark does not extensively evaluate all desired features of a pre-trained model for earth monitoring. For example, it does not evaluate its ability to fine-tune temporal data nor perform fusion with other types of data such as text or weather. The spatial coverage of the benchmark covers most continents and improves coverage over individual datasets. However, the spatial coverage could still be largely improved to include a much wider range of countries and biomes. Finally, as pre-trained models become stronger, they will get closer to the theoretical limit of generalization performance, i.e. approaching the aleatoric uncertainty of the dataset. Under such a regime, we expect a bigger overlap between error bars when comparing 2 different models.

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
