# A Extended Results

In this section, we continue the experiment sections to include other results, that were deferred due to space limitations.

## A.1 Benchmark Coverage

In Figure 8, we depict the coverage of the benchmark on the world map. While there are still large uncovered areas such as China, Russia, North Africa, and South America, the benchmark covers all continents except Antarctica. Most importantly, the coverage of the benchmark is greater than that of individual datasets.

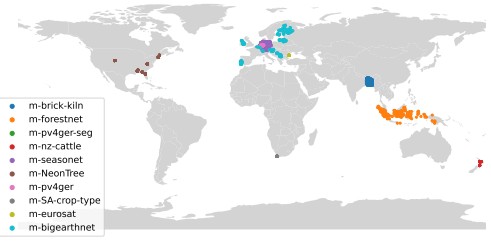

Figure 8: World coverage of the different datasets.

## A.2 Classification

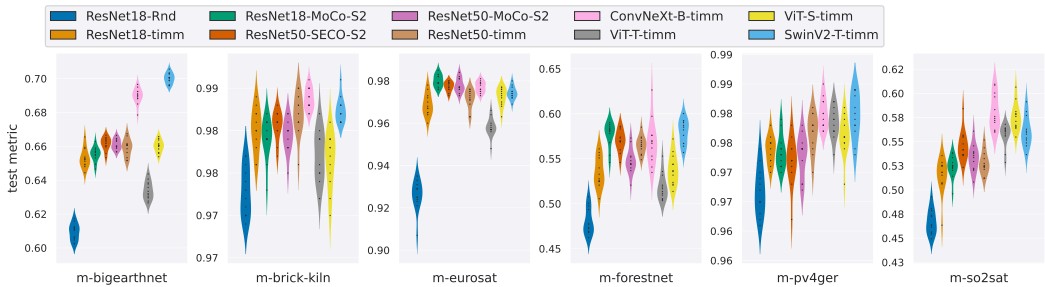

Figure 9: **Classification Benchmark:** Raw accuracies of all seeds of various baselines (higher is better). Violin plots represents the distribution of seeds.

In Figure 9, we report the raw accuracies, before normalisation and IQM. This different perspective, gives a sense of the variance of the results as well as how close it is to the maximum. We recall that uncertainty of the mean expressed in Figure 4 is lower than the variance of the results in Figure 9. This follows from the central limit theorem.

## A.3 Segmentation

In this section, we report results for six baselines on the Segmentation benchmark. First, we introduce the baselines that are evaluated.

### A.3.1 Baselines

**ResNet18 U-Net - ResNet101 U-Net**    ResNet augmented with the U-Net architecture [58] with pre-trained weights from the timm library.

**ResNet18 DeepLabV3 - ResNet101 DeepLabV3**    ResNet augmented with the DeepLabV3 architecture [10] with pre-trained weights from the timm library.

### A.3.2 Comparing Baselines on RGB only

In Figure 10, we report the bootstrapped IQM of the normalized Intersection over Union (IoU) (Sec. 4.1) for the 6 segmentation datasets. In Figure 11, we report the seeds for the 10 experiments.

From the results, we can observe a stronger performance with U-Net architecture and the larger backbone ResNet101 performs a bit less than ResNet50 in general but has less stable performance, leading to slightly lower performance than ResNet50 backbone.

In Figure 12, we observe the behaviour of the baselines as the size of the training set grows from 1% to 100%.

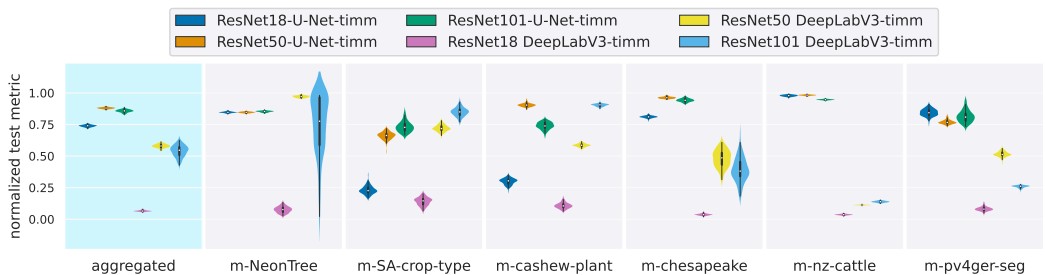

Figure 10: **Segmentation Benchmark:** Normalised intersection over union (IoU) of various baselines (higher is better). Violin plots are obtained from bootstrap samples of normalized IQM (Section 4.1). Left plot reports average across all tasks.

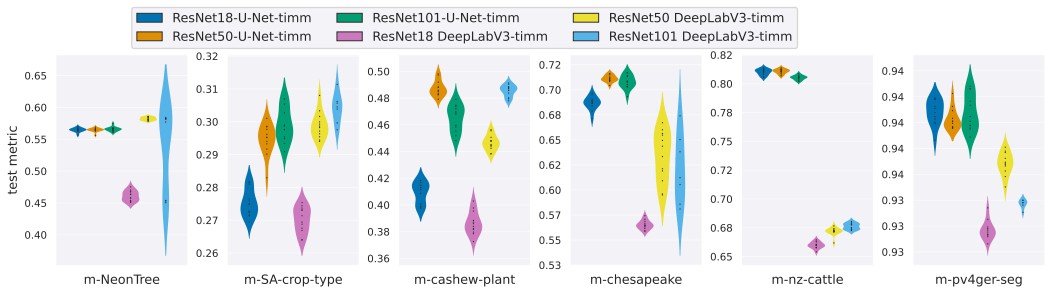

Figure 11: **Segmentation Benchmark:** Raw IoU of all seeds of various baselines (higher is better). Violin plots represents the distribution of seeds.

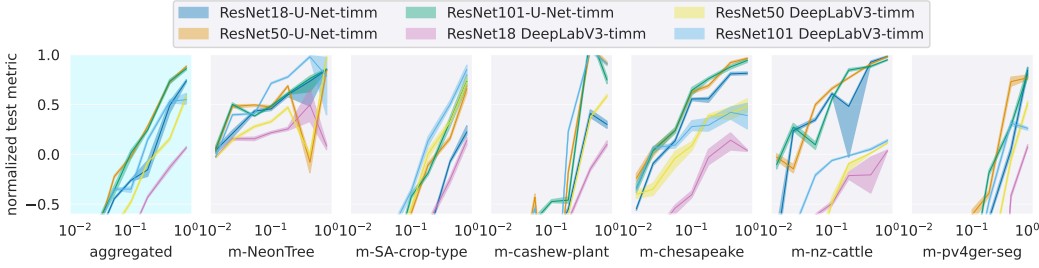

Figure 12: **Segmentation vs Train Size:** Normalised IoU (higher is better) on the Segmentation benchmark with a growing size of the training set. Shaded region represents 80% confidence interval, obtained from bootstrap samples of normalized IQM (Section 4.1). Left plot reports average across all tasks.

### A.4 Convergence Time

As seen in Section 6.2.3, conducting experiments with a growing training set size brings a different and important perspective on the performances of the models. In our experiments, this comes with a seven

fold increase in *number* of experiments. On the surface, this may seem like an excessive amount of computation, but most experiments will run much faster with smaller training set. Indeed, if we decrease the training size at an exponential pace, and we assume that the training time is proportional to the size of the training set, we get a more modest increase in computational need. In GEO-Bench, we generate pre-defined subsets using the following ratios of the training set: ($1\times, 0.5\times, 0.2\times, 0.1\times, 0.05\times, 0.02\times, 0.01\times$). With the proportional training time assumption, this leads to a cumulative $1.88\times$, which is less costly than repeating the experiment twice, and far from the seven fold increase that one could assume.

To confirm the assumption that the training time is proportional to size of the training set, we conduct the following experiment. As a measure of the training time, we use the convergence time i.e., how many training steps[12] are required to achieve the peak performance on the validation set. Let $\tau_{i,j,k}^r$ be the convergence time of model $i$, on dataset $j$, with hyperparameter trial $k$, trained with a ratio $r$ of the training set. Let the reference convergence time be defined as the average convergence time when using 10% of the training set:

$$T_{i,j} = \frac{1}{n_k}\sum_{k=1}^{n_k}\tau_{i,j,k}^{0.1}.$$

Then, the average relative convergence time is:

$$\rho_j^r = \frac{1}{n_i,n_k}\sum_{k=1}^{n_k}\sum_{i=1}^{n_i}\frac{\tau_{i,j,k}^r}{T_{i,j}}.$$

In Figure 13, we plot $\rho_j^r$ for all datasets and all partition size on a log-log plot. We can observe a mild behaviour change near 1% of the training size, but overall we can conclude that the convergence time is proportional the training set size.

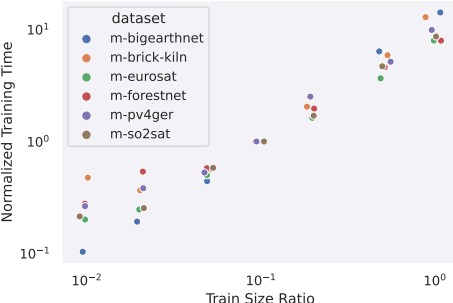

Figure 13: **Convergence Time:** Average time for the training to reach convergence as the training size increase.

## A.5  Discriminativity of Datasets

It is common knowledge in machine learning that larger datasets yields better generalization performances. However, when comes the time to compare algorithms against each other, we hypothesises that less is more i.e. smaller datasets are more likely to exhibit statistical difference between a given pair of models.

In this section we provide experimental evidences on this question. Let $p(A_i^l > A_j^l)$ be the probability that algorithm $i$ is better than algorithm $j$ on dataset $l$. More concretely, $A_i^l$ and $A_j^l$ are random variables corresponding to the accuracies on the validation set of task $l$. We estimate this probability by repeating the training procedure for 10 random seeds, as recommended in Section 4.1, and comparing all pair of results. Using this quantity, we propose the following discriminativity metric

$$d_{ij}^l := 1 - \mathbb{H}_2[p(A_i^l > A_j^l)],$$

where $\mathbb{H}_2$ corresponds to entropy in bits i.e., using $log_2$. This measures how good dataset $l$ is good at telling if algorithm $i$ is better than algorithm $j$. For example, if algorithm $i$ is always better than $j$ or vice

---

[12]We also multiply by the batch size

versa, then we have: $d_{ij}^l = 1$. On the other end if they perform equally well, then $d_{ij}^l = 0$. Next, to get an estimate at the dataset level, we average discriminativity across all pairs of $m$ models to obtain

$$D_l := \frac{1}{m^2} \sum_{ij} d_{ij}^l.$$

Finally, to obtain en estimate of the uncertainty of this measure, we use stratified bootstrap of all experiments, and repeat this procedure 100 times.

In this experiment, we consider a smaller training set as a different $D_l$ value. Hence, with seven different partitions on the 6 tasks of GEO-Bench classification, this leads to a total of 42 different datasets. In Figure 14, we analyse the influence of the training set size on the discriminativity of a dataset, and we conclude:

- Reducing the dataset size almost systematically improve its ability to discriminate between models, but only by a small value.
- BigEarthNet is the most discriminative dataset
- The full size of pv4ger offers poor discriminativity but could be improved if reduced.
- Our estimation of discriminativity is quite noisy and sensitive to the random seeds.

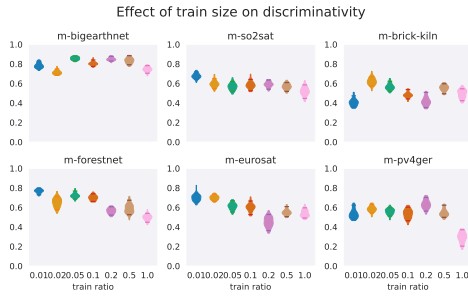

Figure 14: **Discriminativity of datasets:** We report how discriminative are each datasets as we vary the training set size.

### A.6 Resources Usage

For convenience, we also provide resource usage of the different models in Figure 15.

**Number of Parameters:** The number of parameters of a model gives a hint on the overall memory usage, but most importantly on the learning capacity of the model. In Figure 15-left, we see that ConvNeXt is by far the one with the highest capacity. While ViT-T has less parameters than ResNet18. We note that SwinV2 has up to 3 billion parameters with SwinV2-G, but we focus on models that could be run on a single 32 GB GPUs without extra work.

**Memory Usage:** While the number of parameters gives a hint about the memory usage, to capture the memory usage of all hidden states we need to measure the memory usage in action with `nvidia-smi`. This is reported in the second plot of Figure 15.

**Forward time:** We measure the time for a forward pass of the network with a batch size of 32. This gives a sense of the efficiency of the network deployed in production. Values are reported in the third plot of Figure 15. Violin-plot report the distribution of several measurements during one epoch of training on BigEarthNet, and the average value is reported as a solid line.

**Convergence time:** In Figure 15 right, we report the number of training step required to reach peak generalization performance on validation. These values are reported on from the 100% training set, and the violin-plot report the distribution of values across the different tasks of GEO-Bench-classification and the different trials of the hyperparameter search.

## B Remote Sensing Data Schema

**Band** Earth monitoring data comes with a challenging amount of heterogeneity. Fortunately, the transformer architecture offers the opportunity to mix various modalities using encodings such as temporal

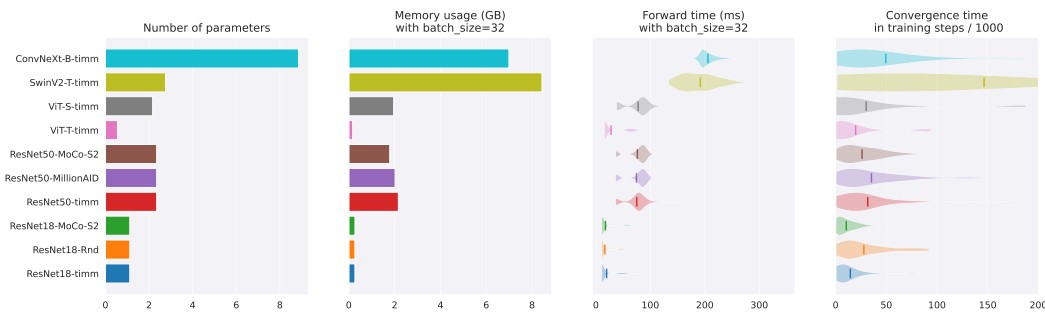

Figure 15: **Resources Usage:** Resources reported for various models. Violin-plots report distribution over several tests and its average as a solid line. See text for more details.

encoding [68] and positional encoding [19]. Similarly, band encoding can be leveraged to communicate the source of the data. To this end, we define `Band` as the core class in our schema. It consists of an array of data with spatial extent accompanied with `BandInfo`, providing information such as the band name, spatial resolution, and spectral range. Through a hierarchy of classes, we also provide the type of sensors (see Figure 16). This provides further information for introspection and flexible information for users to define a *Band Encoding* that could be required for transformer architectures. Finally, a `Sample` is a set of `Band` accompanied by a label which can also be a `Band`.

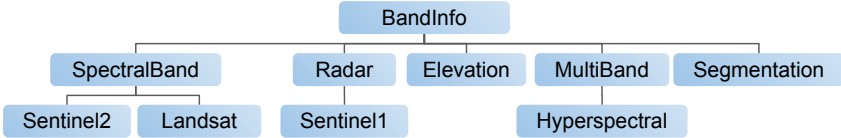

Figure 16: Class hierarchy of `BandInfo` enabling introspection on the type of data.

**Task Specifications**    Each dataset is accompanied by a `TaskSpecifications` object describing the schema of a particular dataset without having to load any samples. It contains the dataset name, the type of labels, the `BandInfo` of each band and their shapes. The aim of this data structure is to let users procedurally generate a machine learning model that is suitable for the given dataset at the beginning of the training.

**Band Statistics**    We also provide band statistics (minimum, maximum, mean, variance, and percentiles). This lets users transform the input data in various possible ways to fit the statistics expected by the pre-trained model.

## C   Societal Impact of Foundation Models for Earth Monitoring

Remote sensing and Earth monitoring have been transformational in the past decades. Applications include military, insurance, market forecasting, climate science, and more. Much of this impact is not directly attributed to deep learning nor large pre-trained networks and its review extends beyond the scope of this section. In this section, our focus is on the impact of bringing foundation models to Earth monitoring.

### C.1   Climate mitigation and adaptation

Machine learning on remote sensing data is widely used to develop solutions for a variety of problems relevant to climate change [8, 57, 77, 45]. The vast majority of these solutions are built by curating datasets for a specific task and require significant resources to develop. Furthermore, the solutions are often tailored to specific regions as extending approaches to new geographies remains a significant challenge, primarily due to the lack of labeled data [77]. Less-economically developed regions of the world are no less susceptible to the impacts of climate change, yet suffer from the lack of effective remote sensing-based solutions [8]. Foundation models for Earth monitoring have the potential to address many of these issues and substantially accelerate and enable the development of new remote sensing solutions for climate change.

## C.2 Increased accessibility

Reducing the need for curating a large labeled dataset for each task could democratise access to the development of machine learning models for remote sensing, specifically for groups or organisations with limited budgets [47, 3]. In particular, foundation models may especially benefit non-profit organisations, academic universities, startups, and developing countries. It may also open opportunities for applications that were not previously profitable. Although we believe that increased accessibility to these models will have a largely net positive impact, we acknowledge that this accessibility may lead to unexpected applications with potentially negative impacts [6]. We also note that such models may have dual-use applications, where, for example, they may help oil and gas industries in their operations in ways that increase (or reduce) overall emissions.

## C.3 Emissions of large pre-trained models

Recent work has investigated emissions of large neural networks [63, 60, 59, 35, 52]. Specifically, training a large transformer can emit 284 $tCO_2e$ when trained on computers using largely fossil fuel energy (US national average) [63]. When put in perspective with individual actions, such emissions are large—e.g., a roundtrip passenger flight from San Francisco to London is 2.8 $tCO_2e$ , about $100\times$ smaller. However, the extensive reusability of pre-trained models and their potential for helping efforts to mitigate climate change [57] calls for a different perspective.

When evaluating new tools and systems, it is important to consider the likely net impact on emissions of both the creation and testing of the tool and its eventual deployment. For example, evaluating the performance of airborne methane sensing tools at emission levels commonly found in oil and gas operations can emit about 7 metric tonnes of methane, roughly 600 $tCO_2e$ equivalent using a 20-year global warming potential [22]. However, in a single day of flying, such a single instrument can survey hundreds of sites, often identifying leaks for repair that emit well over 7 metric tonnes of methane per day [31]. Similarly, foundation models may significantly advance our ability to leverage enormous quantities of passively collected satellite data to massively reduce emissions, qualitatively advance our understanding of climate science, or improve our ability to adapt to climate change.

In sum, the potential benefits for climate change mitigation with improved Earth monitoring methods likely outweigh the emissions associated with foundation models. Moreover, various actions can be taken to reduce and mitigate emissions related to the training of your model [35]:

- Select data centers that are certified carbon neutral or largely powered by renewable energy, with good power usage effectiveness (PUE). Such measures can reduce emissions dramatically $50\times$ reduction in emissions [35].

- Design your code development pipeline to minimize the number of computationally-intensive runs required, e.g. employ modular development and testing when possible.

- Make your code more efficient and sparsify your network when possible [52]. This can reduce emissions up to 10-fold.

- Favour more energy-efficient hardware, e.g., TPUs or GPUs.

- Monitor [59] and report your emissions [35]. Better communication about climate change is fundamental for systemic changes. Better documentation will help other coders pick up where you left off, potentially bypassing some computationally intensive runs.

- Offset the cumulative emissions of your projects.

## C.4 Fairness and biases

Large language models are known to amplify and perpetuate biases [5]. While this can lead to serious societal issues, we believe that biases in remote sensing models are likely to have much less impact. We do however anticipate potential biases and fairness issues.

**Data coverage and resolution**    Some satellites cover the whole Earth with standard spatial resolution and revisit rate (e.g., Sentinel-2 covers the whole Earth at 10-60 m/pixel resolution every 5 days). This makes imagery freely available uniformly across the planet. Other satellite data providers such as Maxar acquire images on-demand and have higher spatial resolution (up to 0.3m per pixel), but also have lower revisit rates and high costs. Some countries, such as New Zealand, freely provide aerial imagery with

resolution up to 0.1m per pixel[13]. Finally, it is worth noting that cloudy seasons in some climates may limit data availability for some countries. Overall, while the coverage is fairly uniform, some regions have much higher coverage than others and money can be a limiting factor to access the data. This can lead to some level of biases and fairness issues.

[13]https://data.linz.govt.nz/