# OpenReview forum: "GEO-Bench: Toward Foundation Models for Earth Monitoring"
_NeurIPS.cc/2023/Track/Datasets_and_Benchmarks — NeurIPS 2023 Datasets and Benchmarks Poster_

### Official Review · Reviewer_xuaP · 2023-07-03
**Review on Geo-bench**

**Rating:** 7
**Confidence:** 4
**Clarity:** Yes, very well written.

**Strengths:**

The work is significantly relevant to many practitioners using earth observation remote sensing data to conduct research or even day-to-day work, including environmentalists, energy sector researchers, social science researcher,s etc.. The quality of the work is also very high as it not only provides great details of the benchmark and experimental results but also gave specific instructions for future users to store their results in csv and how to visualize them, which is rarely seen in other benchmark paper and frankly should be widely adopted. As this work provides great insight and opportunity for a lot of social work (using geospatial data), I think it has a positive social impact overall. Also, the authors limited the replication of their benchmark to single GPU usage, which is great for preventing this research from only benefiting the larger corporations and enabling smaller labs from less economically developed countries to build on this work and potentially benefit the local community.

**Additional Feedback:**

Please fix the tool box documentation as all-around documentation is important for the community to make better use of the work.

**Correctness:**

The claims made in this submission is presented and constructed in a sound way. The evaluation methods and experiment design are appropriate.

**Documentation:**

Folliwing the link of the toolbox (https://github.com/ServiceNow/geo-bench), although the repo has 1000+ commit records, I found little or no documentation in the README file, which is a bit disappointing.

**Limitations:**

Yes the authors addressed the limitation and potential negative social impact of their work.

**Opportunities For Improvement:**

I think there are some opportunities for improvement in the future, for example, adding some parts discussing the pre-training step for the presented baseline, and adding preliminary results for some current "foundation model" benchmark results. Another clear opportunity for improvement would be adding more diverse datasets representing a larger remote sensing community (like UAV / aerial imagery etc.)

**Relation To Prior Work:**

The paper clearly discussed related work and its difference from previous contributions.

**Summary And Contributions:**

The authors proposed a collected of six classification tasks and six segmentation tasks for a variety of earth monitoring application to the community as a benchmark for foundation models. Apart from outlining great details of protocols and metrics to report for future benchmark users, the authors also provided insights into which pre-trained models work best by comparing 20 existing baseline models.

---

> ### Author Response · Authors · 2023-08-11
> **Main response**
>
> We are grateful to have been assigned a reviewer that went in-depth into the details of the paper and provided useful comments that will enhance our work. It's especially encouraging to learn that our efforts found resonance with the reviewer.
>
> ## GitHub README page
> This is a good point. We have been focusing our effort on the paper and experiments up to now. We definitely want to optimize the user experience of this product. Hence, we are working on enhancing the GitHub readme and many other aspects of the code before publicising the benchmark. This should be finalized in the coming few weeks.
>
> ## More diverse dataset
> NeonTree from the segmentation benchmark is a UAV dataset. That said, we agree that the benchmark would benefit from a wider range of UAV datasets. We are planning GEO-Bench 2.0 and we will certainly extend our UAV coverage. Is there a specific dataset that you would recommend?
>
> ## Results from foundation models
> At the time we ran experiments, very few foundation models were available. We found Millionaid, which we report results in section 6. In the past months, a few large models came out already. We are eager to see their performance, we will reach out to authors to encourage them to use our benchmark. We are also planning to set up a leaderboard on papers with code.
>
> ## Pretraining steps
> Could you clarify? You would like us to elaborate on how the baselines were trained before we fine-tuned them? Now that we have an extra page for the camera ready, we can certainly expand on these details.

---

> > ### Comment · Reviewer_xuaP · 2023-08-16
> >
> > The authors do have a lot of time preparing for this rebuttal, while they still have a near-empty github README page.... This is very concerning to me as to me one of the most important part of a benchmark paper would be reproducibility and hence REAMDE is the first step of that.
> >
> > Also for the pertaining steps it would be great if such discussion to "expand on these details" can be part of the rebuttal.
> >
> > Based on the rebuttal content, I would like to lower my score accordingly.

---

> > > ### Author Response · Authors · 2023-08-18
> > > **Re Readme**
> > >
> > > We apologise for the misunderstanding in the expectations. Our initial plan was to first go through the review process and then finalize the benchmark to leave room for potential reviewer requests. Our benchmark is still in version 0.9.0 and we will release version 1.0 before the end of the discussion period. As requested, we’ve updated the readme to highlight the detailed usage. We are in the process of adding more unit tests, improving the documentation and ironing out the user experience.
> > >
> > > We sincerely hope that the reviewer will reconsider the score change. We replied to the rebuttal rapidly after coming back from vacation and had very little time to update the readme.

---

### Official Review · Reviewer_2NRk · 2023-07-20
**One of a kind large-scale dataset for remote sensing image classification and segmentation**

**Rating:** 7
**Confidence:** 3
**Clarity:** Yes, the paper reads well to me.

**Strengths:**

The dataset has a variety of image sizes, image classes, and bands for the classification datasets. Each classification dataset has over 6K training samples, around 1000 validation, and 1000 testing samples. The dataset has fewer images, image classes for the segmentation datasets. Each segmentation dataset has a few hundred to 3K training samples, and around tens to 1000 validation and testing samples. The baseline experiments are generally complete.

**Additional Feedback:**

No more additional feedback for now.

**Correctness:**

The benchmark algorithms and trained models should be open-sourced in order to reproduce the results in the paper, though ResNet, ViT, and others are open-source pre-trained models.

**Documentation:**

There is sufficient detail on data collection and organization. The data is licensed for public usage.

**Ethics:**

There is no ethical concern found in the paper if the images are properly licensed as stated in the paper.

**Limitations:**

Discuss more about the pros and cons of the datasets and the baseline methods.

**Opportunities For Improvement:**

Is there a possibility of training a unified large model that takes care of all six image classification/segmentation tasks as one?

Will you be able to use a new model or network architecture that better performs on your dataset? The new model can be a much better baseline model and do great contributions to the community.

Are there any images that cover the other regions of the Earth?

**Relation To Prior Work:**

There is a full section of related works.

**Summary And Contributions:**

The authors develop an open-source platform containing twelve datasets for remote sensing image classification and segmentation tasks, six datasets for each task. The images have different sizes, from 32*32 to 256*256 for the classification task, and from 256*256 to 500*500 for the segmentation task. All the images are licensed for permission of use.

And the authors show experiments on using several pre-trained models, including ResNet and ViT, as the baseline models and transfer the trained model knowledge by adding an extra randomly initialized layer to adapt to the specific tasks.

Through this benchmark, the authors try to solve the questions:
"• Which new architecture performs best on remote sensing data (Section 6.2.2)?
• What is the effect of training set size on the performance of each model (Section 6.2.3)?
• Can we leverage multispectral channels to improve performance (Section 6.2.4)?
• Are smaller datasets better at discriminating the performance of different models (Section A.5)?"
Which I think are important for the data-driven remote sensing community.

---

> ### Author Response · Authors · 2023-08-11
> **Main Response**
>
> We note that this paper is not about developing new models for earth monitoring but focuses on developing a reliable tool for measuring existing models and future progress.
>
> Fundamentally, we care about the ecosystem of tools that will generate great foundation models for Earth monitoring and we hope this will play a signifanct role in tackling climate change.
>
>
> ## Unified Model
> Yes it would definitely be possible to train a single model capable of performing on all 12 datasets of the benchmark. It could have two heads for segmentation and classification. The set of classes would be the union of all classes and it would have to be trained in a multi-label fashion since a single image could contain multiple classes. Our benchmark would be a good tool for evaluating the performance of such a model. However, performing this experiment goes beyond the scope of this paper.
>
> ## New model with better performances
> As a separate project we are working on developing new models that outperform existing models on this Earth monitoring benchmark. We are also aware of other groups working on developing new models and we expect a large variety of models to make their apparition in the coming years. We hope that GEO-Bench will be a valuable tool for comparing these models and measuring progress in a fair and reliable way.
>
> ## Data coverage
> This is an important point. For many datasets, the georeferencing information was tedious to access and properly convert. After plotting the location of the samples we were surprised to observe a somewhat limited coverage of the benchmark given the wide range of tasks. This is something we would like to improve in GEO-Bench 2.0. We will add this point to the limitation section.
>
> ## More pros and cons
> We will extend the limitation section. Is there anything specific you would like to be discussed?
>
> ## should be open-sourced
> Everything is open-sourced under very flexible licenses, code for reproducing experiments will also be open-sourced.

---

### Official Review · Reviewer_HGjk · 2023-07-20
**GEO-Bench: Toward Foundation Models for Earth Monitoring**

**Rating:** 9
**Confidence:** 3
**Correctness:** Yes
**Clarity:** Clear

**Strengths:**

GEO-Bench encompasses 6 classification tasks and 6 segmentation tasks.
The paper is novel, well written and does an ample set of analyses.

**Additional Feedback:**

No additional feedback at this point.

**Documentation:**

This could be improved by providing a step-by-step of how to use GEO-Bench and how to get set-up, perhaps in their GitHub repo

**Limitations:**

I would add data coverage as another limitation

**Opportunities For Improvement:**

A few suggestions for improvement:
1. While the authors compare benchmarks in section 6.2.2. I would appreciate a section discussing how this could be done on other EO data so moving beyond RGB and would those results hold beyond RGB data?
2. As a new user of GEO-Bench, where should one start? A step-by-step guide on their GitHub page would be very helpful
3. the 6 classification. and 6 segmentation tasks are related to terrestrial EO challenges primarily, what about including a dataset that relates to terrestrial water resources?
4. the data coverage is pretty aggregated in some areas of the globe while others are not covered at all (South America) - how do the authors propose to handle this shortcoming? Perhaps this point should be added under limitations

**Relation To Prior Work:**

Yes, the authors do a good job summarizing prior work

**Summary And Contributions:**

This paper addresses the gap of foundation models for earth observations (EO), with the authors proposing GEO-Bench. While there computer vision specific foundation models (e.g. CLIP, DINO) they do not perform well on EO data as they are collected differently (e.g. ground point vs satellite orbits) and trained on different kinds of images (e.g. RGB vs multi-spectral or radar bands).

---

> ### Author Response · Authors · 2023-08-11
> **Main response**
>
>
> We are grateful to have been assigned a reviewer that went in-depth into the details of the paper and provided useful comments that will enhance our work. It's especially encouraging to learn that our efforts found resonance with the reviewer.
>
> ## Beyond RGB
> In section 6.2.4, we discuss experiments on the multi-spectral part of the benchmark. Figures 6 and 7 report the results. Perhaps we misunderstood the question. If so, could you clarify?
>
> ## GitHub page
> Absolutely. We have been focusing our effort on the paper and experiments up to now. We definitely want to optimize the user experience of this product. Hence, we are working on enhancing the GitHub readme and many other aspects of the code before publicising the benchmark. This should be finalized in the coming few weeks.
>
> ## Terrestrial water resources
> This would be great. We are planning GEO-Bench 2.0 and will add a wider range of datasets. Could you recommend a dataset for water resources?
>
> ## Data coverage
> This is an important point. For many datasets, the georeferencing information was tedious to access and properly convert. After plotting the location of the samples we were surprised to observe a somewhat limited coverage of the benchmark given the wide range of tasks. This is something we would like to improve in GEO-Bench 2.0. We will add this point to the limitation section.

---

### Official Review · Reviewer_PQwP · 2023-07-21
**First Review of GEO-Bench: Toward Foundation Models for Earth Monitoring**

**Rating:** 10
**Confidence:** 3
**Clarity:** The paper is written in a very concis…

**Strengths:**

The submission provides a very comprehensive benchmark covering a multitude of different model architectures as well as a variety of downstream tasks supported by the benchmark datasets. Its contribution to the broader research community is highly significant and timely considering both the rise of foundation models as the new AI paradigm and the urgency of Earth monitoring technology in the face of climate change.

In addition to that, the paper is very well structured and describes clearly and transparently the principles of evaluation and the experimental results which demonstrates the high quality of the benchmark and the underlying research.

The supplementary material provide additional insightful details on baseline results, dataset characteristics and resource usage. In addition, ethical and social implications are comprehensively discussed.


**Additional Feedback:**

L90: Figure 3 should be referenced before Figure 4.

Appendix, footnote 13: “..to reduce the frequency of failure” instead of “..to reduce to frequency of failure”


**Correctness:**

The benchmark is designed in an appropriate way and the evaluation of the baseline models follow a strict protocol. This allows this benchmark to be a fair point of comparison for new models.

**Documentation:**

Benchmark datasets are properly archived with a persistent dereferenceable identifier (DOI).
The paper contains clear instructions on how to use the benchmarks and how to report new results.
Instructions on the associated GitHub repo on how to get started are “coming soon” and are hopefully added by the time of publication.



**Ethics:**

No ethical concerns are observed.

**Limitations:**

The authors reflect on limitations of the presented benchmark with regards to limited dataset modalities and model generalization power and discuss comprehensively on the potential societal impacts from a variety of angles.

**Opportunities For Improvement:**

The work is comprehensive and provides a clear benchmark with a well described experimenting and reporting protocol.
Documentation on the associated GitHub page should be improved to allow other researchers an easier start to using the benchmark.


**Relation To Prior Work:**

The submission includes a section on related works which clearly differentiates the present benchmark from prior works. All baseline models are properly referenced, as is the literature review in the introductory paragraph.

**Summary And Contributions:**

The authors present a new benchmark for the evaluation of foundation models from remote sensing data for applications related to Earth monitoring. The construction of this benchmark is motivated by providing a brief literature review and types of remote sensing data are introduced. The design principles of the benchmark, which consists of six classification tasks and six segmentation tasks, are reported as well as how the benchmark datasets are constructed. Instructions are given of how the benchmark is to be used and how results are to be reported. Results of 20 baseline models are reported and discussed to answer fundamental questions with regards to foundation models for remote sensing data.

---

> ### Author Response · Authors · 2023-08-11
> **Main response**
>
> We are grateful to have been assigned a reviewer that went in-depth into the details of the paper and provided useful comments that will enhance our work. It's especially encouraging to learn that our efforts found resonance with the reviewer.
>
> ## GitHub page
> This is a good point. We have been focusing our effort on the paper and experiments up to now. We definitely want to optimize the user experience of this product. Hence, we are working on enhancing the GitHub readme and many other aspects of the code before publicising the benchmark. This should be finalized in the coming few weeks.
>
> ## Typos
> The typos have been corrected. Thank you for reporting.

---

> > ### Comment · Reviewer_PQwP · 2023-08-30
> > **Response to the authors**
> >
> > Dear authors,
> > Thanks for you comments to my review. I have seen the updated readme which is very much appreciated.
> > I believe the submission is a valuable benchmark for the community.

---

### Official Review · Reviewer_4soD · 2023-07-21
**Necessary grounding for foundational models**

**Rating:** 8
**Confidence:** 4

**Strengths:**

1. A standardized benchmark for evaluating pre-trained models. This allows for consistent comparisons between different models and facilitates the development of foundation models.

2. The open evaluation procedure enables transparency and comparisons across new datasets and new models.

3. A diverse set of image classification and semantic segmentation tasks that are curated by domain experts to ensure they are also relevant, A wide range of Earth monitoring applications can be tested on this benchmark.



**Additional Feedback:**

Please add appendices to the submission.

**Clarity:**

The paper is concise, almost to the point of being terse.  I wouldn't have minded more detail.
But it is very clear and understandable.

**Correctness:**

The datasets and the benchmark are all constructed in a very sound, reasonable way.

**Documentation:**

Yes, there are links to the github repo and datasets which allow for reproducibility

**Ethics:**

For the datasets used here, there shouldn't be any concerns. However, the authors do not explicitly address this. The appendices were not included in this paper.

**Limitations:**

In the conclusion section, the authors list some aspects of pre-trained models that they have not covered.

**Opportunities For Improvement:**

The reporting of results with the IQM instead of the mean strikes this reviewer as potentially discarding valuable information. Higher variances, caused by outliers or by other mechanics, are not necessarily bad, and may result in a different ordering of the models' performance. It would be useful at some future point to test and report on other metrics to see how results differ.

As people use this benchmark, it may not be possible for them to use it exactly as the authors recommend because of various resource constraints. It's not clear how results from new datasets will be added to the existing body of results. However, that is an issue that can be solved over time, and is no reason not to use this benchmark.

Fine-tuning and hyperparameter tuning. The authors might receive requests for clarification on whether they have guidelines on how to conduct tuning, or once done, how to report the tuning.

**Relation To Prior Work:**

The authors briefly describe a couple of existing benchmark proposals. They could have included other recent work such as AiTLAS.

**Summary And Contributions:**

For researchers who will undoubtedly create foundational models, the benchmark in this paper will help them compare their models to existing pre-trained models. It can also be used by researchers who want to know which pre-trained model might perform best for their data.

---

> ### Author Response · Authors · 2023-08-11
> **Main response**
>
> We are grateful to have been assigned a reviewer that went in-depth into the details of the paper and provided useful comments that will enhance our work. It's especially encouraging to learn that our efforts found resonance with the reviewer.
>
> ## IQM discards useful data points.
> Your point is well taken. We will add an option to display the mean within our Python tools. However, it's worth noting that, in practice, models which underperform due to random initialization are typically filtered out using the validation set, which are the main outliers.
>
> ## How to add new datasets:
> In order to be able to compare results between models, it is important to have a static version of the benchmark. Hence it is not possible to add datasets at will. However, we have plans to develop GEO-Bench 2.0 which would include a wider set of datasets and would welcome recommendations from the community.
>
> ## Fine-tuning and hyperparameter tuning:
> While this is not part of the pre-trained model, tuning methodologies and hyperparameter selection are still active research and can have a significant impact on the final performances. For this reason, we leave it as a design choice to the user of the benchmark. Our existing guidelines suggest a uniform budget for hyperparameter trials across datasets to ensure fair play, irrespective of GPU availability. Thanks to your recommendation we will add guidelines on how to report tuning methodologies.
>
> ## The appendices were not included in this paper:
> It is forbidden to provide appendices in the main submission. In supplementary materials, we provide a long version of the paper with appendices.

---

### Decision · Program_Chairs · 2023-09-22

**Decision:**

Accept (Poster)

**Comment:**

This is a timely paper and aligns well with the rise of foundation models and the need for strategies to process wide amounts of remote sensing data. Indeed, it relies on existing data, but I think it is a great example of things we can do with remote sensing and I am happy to recommend acceptance.

The authors have replied to all questions and convinced the reviewers.